# Learning Universal User Representations via Self-Supervised Lifelong Behaviors Modeling

## Abstract

Universal user representation is an important research topic in industry, and is widely used in diverse downstream user analysis tasks, such as user profiling and user preference prediction. With the rapid development of Internet service platforms, extremely long user behavior sequences have been accumulated. However, existing researches have little ability to model universal user representation based on lifelong sequences of user behavior since registration. In this study, we propose a novel framework called Lifelong User Representation Model (LURM) to tackle this challenge. Specifically, LURM consists of two cascaded sub-models: (i) Bag of Interests (BoI) encodes user behaviors in any time period into a sparse vector with super-high dimension (*e.g.*, $10^5$); (ii) Self-supervised Multi-anchor Encoder Network (SMEN) maps sequences of BoI features to multiple low-dimensional user representations by contrastive learning. SMEN achieves almost lossless dimensionality reduction, benefiting from a novel multi-anchor module which can learn different aspects of user preferences. Experiments on several benchmark datasets show that our approach outperforms state-of-the-art unsupervised representation methods in downstream tasks.

## 1 Introduction

Inferring user intents through user behavior data has been extensively studied in industrial applications, such as recommendation systems, search engines, and online advertising Dupret & Piwowarski (2008); He et al. (2014); Elkahky et al. (2015); Yu et al. (2016). One key aspect in these systems is user modeling, which describes the process of building up and modifying a conceptual understanding of the user Fischer (2001). Essentially, user modeling is to learn a user representation that helps to capture the user's interests and preferences to improve the performance on downstream tasks.

In the literature, there have been many studies that focused on task-specific user modeling, such as user response prediction and personalized recommendation Ren et al. (2019); Yu et al. (2019); Ji et al. (2020). However, the user representation learned by a specific task can hardly be generalized to other tasks. As a result, specific user representation models need to be trained in each downstream task, which requires massive labeled data, training time, computing and storage resources. Given these limitations, universal user representations that can serve a variety of downstream tasks are preferred.

Due to the sequential form of user behavior data, recurrent neural networks (RNNs) are usually used to encode the temporal dynamics of user behavior sequences Wu et al. (2017); Devooght & Bersini (2017); Zhu et al. (2017); An et al. (2019). Unfortunately, these approaches can only process user behavior sequences with a length of tens or hundreds, while the length can reach hundreds of thousands in many social networks and e-commerce services. Moreover, we have verified by experiments that the performance will become better as the user behavior becomes richer in most downstream tasks (Fig. 3). To solve this problem, many methods drown from natural language processing (NLP) Kumar et al. (2016); Dai et al. (2019); Yang et al. (2019) are proposed. They model long sequential data based on hierarchical architectures and memory networks Ying et al. (2018); Pi et al. (2019); Ren et al. (2019). However, it is still hard for them to encode lifelong user behavior sequences when the length scales up to 1000. And representations they learned through a specific task result in poor generalization capabilities.

In this work, we propose a novel framework called Lifelong User Representation Model (LURM) to model user behaviors since registration. To meet the needs of extremely long sequence modeling, we first introduce a model named Bag of Interests (BoI) to summarize items in behavior sequences similar to Bag of Visual Words. In this way, we can use a sparse vector with super-high dimension to represent user behavior in any time period. Then, a Self-supervised Multi-anchor Encoder Network (SMEN) that maps sequences of BoI features to multiple low-dimensional user representations is proposed. SMEN consists of three modules: a multi-anchor module which can learn different aspects of user preferences, a time aggregation module which can model evolution of user behaviors, and a multi-scale aggregation module which can learn and aggregate BoI features in different scales. Considering the consistency between user behaviors in different time periods, we introduce a contrastive loss function to the self-supervised training of SMEN. With the designs above, SMEN achieves almost lossless dimensionality reduction.

The main contribution of our work can be summarized as follows:

- In this work, a novel framework named LURM is proposed to model lifelong user behaviors of any length. To the best of our knowledge, it is the first method that has the ability to model lifelong behaviors in the field of universal user representation learning.

- We introduce a sub-model named BoI which can encode behaviors in any time period, so that lifelong behavior data can be represented by a sequence of sparse vectors.

- We can obtain compressed user representations with little information loss with the help of a designed sub-model named SMEN.

- Extensive experiments are performed on several real-world datasets. The results demonstrate the effectiveness and generalization ability of the learned user representation.

## 2 RELATED WORKS

### 2.1 UNIVERSAL USER MODELING

Compared with task-specific user modeling that requires more resources, universal user representations are preferred to serve different downstream tasks. In recent years, some works dedicated to learning universal user representations have been proposed. Ni et al. (2018) proposed a representation learning method based on multi-task learning, which enabled the network to generalize universal user representations. Extensive experiments showed the generality and transferability of the user representation. However the effectiveness of this method may still suffer due to the selection of tasks and the need of labels. To release the burden of labeling, Andrews & Bishop (2019) proposed a novel procedure to learn user embedding by using metric learning. They learned a mapping from short episodes of user behaviors to a vector space in which the distance between points captures the similarity of the corresponding users' invariant features. Gu et al. (2020) proposed a network named self-supervised user modeling network (SUMN) to encode user behavior data into universal representation. They introduced a behavior consistency loss, which guided the model to fully identify and preserve valuable user information under a self-supervised learning framework. Wu et al. (2020) proposed pre-trained user models (PTUM), which can learn universal user models based on two self-supervision tasks for pre-training. The first one was masked behavior prediction, which can model the relatedness between historical behaviors. The second one was next K behavior prediction, which can model the relatedness between past and future behaviors. Unfortunately, these methods can only process user behavior sequences with a length of hundreds, and cannot leverage the rich information brought by lifelong user behaviors.

### 2.2 LIFELONG USER MODELING

Previous works have shown that considering long-term historical behavior sequences for user modeling can significantly improve the performance of different tasks Ren et al. (2019); Pi et al. (2019; 2020). Ren *et al.* proposed a hierarchical periodic memory network for lifelong sequential modeling. They built a personalized memorization for each user, which remembers both intrinsic user tastes and multi-facet user interests with the learned while compressed memory. Pi *et al.* decoupled the user modeling from the whole CTR prediction system to tackle the challenge of the storage cost

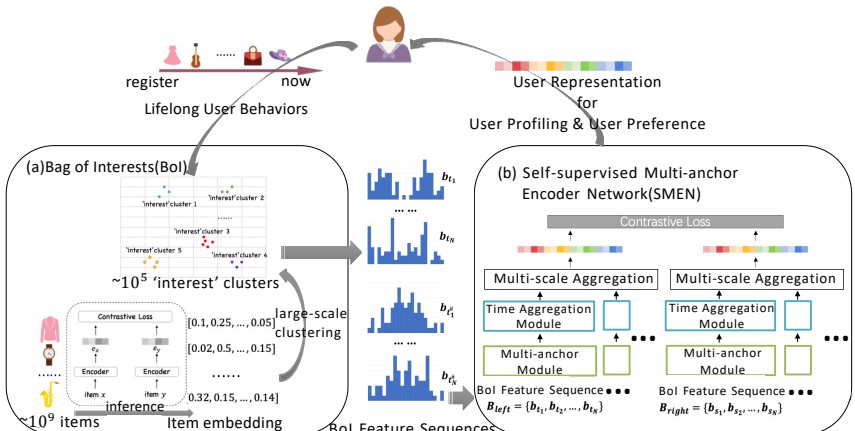

Figure 1: Illustration of our Lifelong User Representation Model (LURM) for user understanding. LURM consists of two sub-model: (a) Bag of Interests (BoI) is used to aggregate user behavior data at 'interest' granularity, and is composed of an item embedding module and a large scale clustering module. We apply the BoI model on lifelong behavior data to generate multi-scale BoI feature sequences, (b) Self-supervised Multi-anchor Encoder Network (SMEN) is used to learn compressed user representations from BoI features sequences, and is composed of a multi-anchor module, a time aggregation module, a multi-scale aggregation module and a contrastive learning module.

and the system latency. Specifically, they proposed a user interest center module for real-time inference and a memory-based network that can be implemented incrementally. Pi *et al.* also designed a search-based interest model (SIM) with a cascaded two-stage search paradigm to capture the diverse long-term interest with target item. Unfortunately, the length of the user behavior sequence that these models can handle is still limited. Moreover, these models are all trained on specific tasks, which limits the generalization ability.

## 3  METHODOLOGY

In this work, we are committed to learning universal user representation through truly lifelong user behavior sequences with arbitrary length. For this purpose, we propose a framework named Lifelong User Representation Model (LURM) which consists of two cascaded sub-models: Bag of Interests (BoI) and Self-supervised Multi-anchor Encoder Network (SMEN). The overall architecture of LURM is shown in Fig. 1.

### 3.1  BAG OF INTERESTS

In order to model extremely long lifelong user behavior sequence, we propose to aggregate the content of items under user purchases, clicks, or other behaviors at a certain granularity. Inspired by Bag of Visual Words (BoVW) Fei-Fei & Perona (2005), item is the natural granularity for aggregation. But there are billions of items on the entire e-commerce platform, which means that the item vocabulary is extremely large, making it infeasible in practice. Therefore, we propose a model called Bag of Interests (BoI) to aggregate user behavior data at 'interest' granularity. Every 'interest' is a cluster of similar items and represents a certain kind of preference. The size of 'interest' vocabulary is often selected at a level of about $10^5$ for retaining enough details. As shown in Fig. 1 (a), BoI consists of an item embedding module and a large-scale clustering module. For convenience, we focus on text modality only in this work. It should be noted that our method can be extended to multi-modal data easily.

#### 3.1.1  ITEM EMBEDDING MODULE

Like the BoVW model, an 'interest' vocabulary are supposed to be built in our BoI model. Therefore, the embedding of each item is required, so that similar items with close distance in the embedding space can be clustered together. Recently, in natural language and image processing, dis-

criminative approaches based on contrastive learning in the latent space have shown great success in the field of representation learning, achieving state-of-the-art results. Inspired by these works, we design a contrastive learning task based on the relation between items drawn from a user to learn item embedding similar to item2vec Barkan & Koenigstein (2016).

Given a set of users $U = \{u_1, u_2, ..., u_{|U|}\}$, each user $u \in U$ corresponds to a behavior sequence $S = \{x_1, x_2, ..., x_{|S|}\}$, where $x_i \in S$ denotes the $i$-th item. $|U|$ and $|S|$ denote the number of users and the length of $u$'s behaviors respectively. Generally, the content of an item $x$ can be expressed as $\{w_1, w_2, ..., w_{|x|}\}$, where $w_i$ denotes a word from a vocabulary $V$, and $|x|$ denotes the number of words in the content of $x$. Firstly, an encoder with average operation is used to generate item embedding $e$:

$$e_x = encoder(w_1, w_2, ..., w_{|x|}) = proj(\frac{1}{|x|}\sum_{i=1}^{|x|} W_i), \tag{1}$$

where $W_i \in \mathbb{R}^d$ is the embedding of word $w_i$ and will be learned during training, $proj(\cdot)$ includes two residual blocks, and a $L_2$ normalization layer. To construct the contrastive learning task, we sample positive pairs from behavior sequences of users randomly. Specifically, two items $(x_i, y_i)$ are similar, i.e. a positive pair, if they are drawn from the same user behavior sequence and the time interval between the occurrence of these two items is less than $\beta$, where $\beta$ is the window size controlling the interval of the two user behaviors. Without loss of generality, the sampled mini-batch with batch size $n$ can be denoted as $\Delta = \{x^1, y^1, x^2, y^2, ..., x^n, y^n\}$, where $(x^i, y^i)$ construct a positive pair drawn from the behavior sequence $S_t^i$ of the $i$-th user in batch. Then, the contrastive prediction task is defined to identify $y^i$ in $\Delta \backslash \{x^i\}$ for a given $x^i$, and all other items in $\Delta \backslash \{x^i, y^i\}$ are negatives. The loss for the positive pair $(x^i, y^i)$ is written as

$$l(x^i, y^i) = -\log \frac{e^{g(x^i, y^i)/\tau}}{\sum_{\nu \in \Delta, \nu \neq x^i} e^{g(x^i, \nu)/\tau}}, \tag{2}$$

where $g(x, y) = \frac{e_x^T e_y}{\|e_x\|\|e_y\|} = e_x^T e_y$ denotes the cosine similarity between the embedding $e_x$ and the embedding $e_y$, and $\tau$ is the temperature parameter. The final objective is the average loss of all positive pairs in the mini-batch, which can be written as

$$Loss = \frac{1}{2n}\sum_i (l(x^i, y^i) + l(y^i, x^i)). \tag{3}$$

Noting that, other methods which can be used to generate item embedding are feasible.

### 3.1.2 LARGE-SCALE CLUSTERING MODULE

After the encoder above has been trained, an item embedding set $E = \{e_i\}_{i \in I}$ is obtained, where $I$ is the complete collection of items at the billion level. In order to retain details as many as possible, the size of the 'interest' vocabulary is set to be at $10^4 \sim 10^5$ level. In other words, all items should be clustered into $D$ (e.g., $10^5$) categories. Considering the large-scale item set, a subset $E' \subset E$ at the million level is sampled, and an efficient clustering algorithm named HDSC Yi et al. (2014) on $E'$ is employed to cluster similar items into the same 'interest'.

After clustering, the cluster centers $C$ make up an 'interest' vocabulary. Therefore each item can be attributed to one/multiple 'interest(s)' by hard/soft cluster assignment. Take hard cluster assignment as an example, each user can obtain his/her sparsely high-dimensional BoI feature $b_t \in \mathbb{R}^D$ in time period $t$ according to his/her behavior sequence $S_t = \{x_1, x_2, ..., x_{|S_t|}\}$ :

$$b_t = [log(1 + \Sigma_{i=1}^{|S_t|}\mathbb{I}_{x_i \in c_1}), log(1 + \Sigma_{i=1}^{|S_t|}\mathbb{I}_{x_i \in c_2}),$$
$$..., log(1 + \Sigma_{i=1}^{|S_t|}\mathbb{I}_{x_i \in c_D})] \tag{4}$$

where $x_i \in c_j$ means item $i$ is assigned to the cluster $c_j \in C$, $\mathbb{I}$ is an indicator function.

### 3.2 SELF-SUPERVISED MULTI-ANCHOR ENCODER NETWORK

As mentioned above, a BoI feature $b_t$ for each user is obtained, through the BoI model given behavior data in any time period $t$. Most directly, a user representation with super-high dimension $b_T$ can

be obtained by applying the BoI model to the whole lifelong time $\boldsymbol{T}$. However, there are two main disadvantages: 1) it is not friendly to the downstream tasks since the dimension of representation is too high, and 2) it is too crude to aggregate the information in the whole lifelong time without considering variations over time.

Therefore, we propose to get a BoI feature sequence by applying the BoI model at each time period of the whole lifelong time. Then, a Self-supervised Multi-anchor Encoder Network (SMEN) is designed to learn compressed user representations from the sequence of BoI features.

The lifelong time $\boldsymbol{T}$ is divided into $N$ parts at a fixed time interval, i.e. $\boldsymbol{T} = \{\boldsymbol{t_1}, \boldsymbol{t_2}, ..., \boldsymbol{t_N}\}$ and $\boldsymbol{t_i}$ denotes the $i$-th time period. In our experiments, the time interval is usually set to be monthly/seasonly/yearly granularity. In this way, a sequence of BoI features $\boldsymbol{B} = \{\boldsymbol{b_{t_1}}, \boldsymbol{b_{t_2}}, ..., \boldsymbol{b_{t_N}}\}$ can be obtained, where $\boldsymbol{b_{t_i}}$ denotes the BoI feature corresponding to the $i$-th time period $\boldsymbol{t_i}$. After that, SMEN is used to map $\boldsymbol{B}$ to low-dimensional user representations. As shown in Fig. 1 (b), SMEN consists of a multi-anchor module, a time aggregation module, a multi-scale aggregation module and a contrastive learning module. Details of the model will be described in the following subsections.

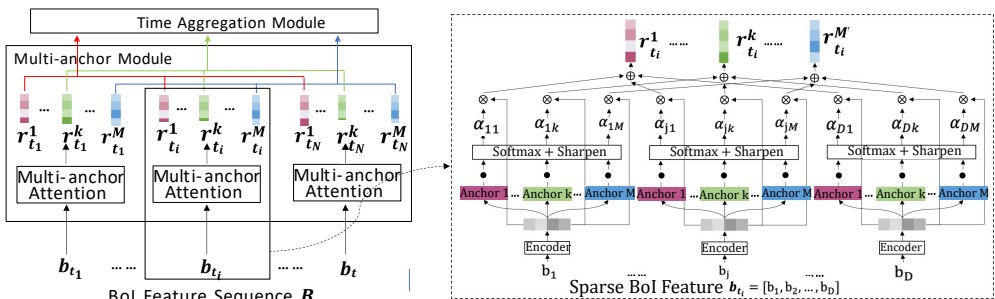

Figure 2: Illustration of the multi-anchor module in SMEN. The module outputs a group of diverse representations by assigning different portions of each 'interest' to different anchors. The ● denotes the dot product operation, ⊗ denotes the scalar multiply operation, and ⊕ denotes vector sum operation.

### 3.2.1 MULTI-ANCHOR MODULE

The data of user behavior, which is highly unstructured and complex, implies different preferences of the user. To capture diverse aspects of user preferences, a novel multi-anchor module is proposed. Specifically, suppose there are $M$ anchors, and each of them indicates a certain preference of users. Let $\boldsymbol{b}$ be a BoI feature, the module converts $\boldsymbol{b}$ to $M$ low-dimensional representations as shown in Fig. 2. Each representation is computed as

$$\boldsymbol{r^i} = ReLU(\boldsymbol{\alpha_i}^T f(\boldsymbol{b})) = ReLU(\sum_j^D \alpha_{ij} f(b_j)) = ReLU(\sum_j^D \alpha_{ij} b_j \boldsymbol{W_j^e}), \quad (5)$$

where $f(b_j) = b_j \boldsymbol{W_j^e}$ is the 'interest' embedding function, $b_j$ is the $j$-th element of $\boldsymbol{b}$, $\boldsymbol{W^e} = (\boldsymbol{W_1^e}, \boldsymbol{W_2^e}, ..., \boldsymbol{W_D^e})^T$ is the embedding matrix. And $\alpha_{ij}$ is the attention weight between the $i$-th anchor and the $j$-th 'interest', which measures the portion assigned to the $i$-th preference from the $j$-th behavior 'interest'. The weight $\alpha_{ij}$ is defined as

$$\alpha_{ij} = \frac{\exp(\boldsymbol{W_i^a} \boldsymbol{k_j})}{\sum_l \exp(\boldsymbol{W_l^a} \boldsymbol{k_j})}, \quad (6)$$

where $\boldsymbol{W_i^a}$ is the anchor vector corresponding to the $i$-th anchor, $\boldsymbol{W^a} = (\boldsymbol{W_1^a}, \boldsymbol{W_2^a}, ..., \boldsymbol{W_M^a})^T$ is the anchor matrix, and $\boldsymbol{k_j} = \boldsymbol{W^p} ReLU(\boldsymbol{W_j^e})$ is the interest vector corresponding to the $j$-th 'interest'. $\boldsymbol{W^e} \in \mathbb{R}^{D \times H}$, $\boldsymbol{W^a} \in \mathbb{R}^{M \times H}$, and $\boldsymbol{W^p} \in \mathbb{R}^{H \times H}$ are learned parameters. $\boldsymbol{r^i}$ can be computed efficiently since $\boldsymbol{b}$ is a sparse vector. Due to the different anchor vectors, different attention weights can be generated for each 'interest'. Finally, a group of different aggregated representations $\boldsymbol{R} = \{\boldsymbol{r^1}, \boldsymbol{r^2}, ..., \boldsymbol{r^M}\}$ can be obtained ($M$ indicates the total number of anchors). In this way, we

can learn different aspects of user preferences. Specially, experiments prove that SMEN can achieve almost lossless dimensionality reduction mainly due to the multi-anchor module.

### 3.2.2 Time Aggregation Module

Through the multi-anchor module, a sequence of representation groups $\mathcal{R} = \{R_{t_1}, R_{t_2}, ..., R_{t_N}\}$ for each user is obtained, where $R_{t_i} = \{r_{t_i}^1, r_{t_i}^2, ..., r_{t_i}^M\}$ is a representation group generated by multi-anchor module corresponding to the BoI feature $b_{t_i}$ in time period $t_i$. In the time aggregation module, each sequence $R^i = \{r_{t_1}^i, r_{t_2}^i, ..., r_{t_N}^i\}$ is aggregated separately to yield a new representation $\tilde{r}^i \in \mathbb{R}^H$. Thus the variable-size representation sequence $\mathcal{R}$ can be transformed into $M$ fixed-size representations $\tilde{R} = \{\tilde{r}^1, \tilde{r}^2, ..., \tilde{r}^M\}$.

There are various methods which can be used to aggregate variable-size sequences, such as average/max pooling and RNNs. Compared to average and max pooling, RNNs are more appropriate to capture variations over time. Among all RNN-based models, long short-term memory (LSTM) and gated recurrent units (GRU) are most commonly used. Considering that GRU has fewer parameters and is less computationally intensive, GRU is adopted to the time aggregation module in this work.

### 3.2.3 Multi-scale Aggregation Module

Under a time division $T = \{t_1, t_2, ..., t_N\}$, we have obtained $M$ representations $\tilde{R} = \{\tilde{r}^1, \tilde{r}^2, ..., \tilde{r}^M\}$ through the BoI, multi-anchor module and time aggregation module as introduced above. It's worth noting that if the time interval is too small, the input sequence of SMEN, i.e. $B = \{b_{t_1}, b_{t_2}, ..., b_{t_N}\}$ will become extremely long, which contains extensive details but causes modeling difficulties due to catastrophic forgetting. On the other hand, details may be lost if the time interval is too large.

To address this trade-off, multi-scale aggregation module is designed. The module captures diverse patterns from user behavior by aggregating several representations generated at different granularities. Generally, user behavior in this work is aggregated monthly and yearly, i.e. $T = \{t_1, t_2, ..., t_N\} = \{t'_1, t'_2, ..., t'_{N'}\}$ (e.g., if the length of time period is 5 years, $N$ is 60 for monthly granularity, and $N'$ is 5 for yearly granularity). Thus two sequences of BoI feature $B = \{b_{t_1}, b_{t_2}, ..., b_{t_N}\}$ and $B' = \{b_{t'_1}, b_{t'_2}, ..., b_{t'_{N'}}\}$ at different scales correspondingly are obtained. Then, two groups of representations $\tilde{R} = \{\tilde{r}^1, \tilde{r}^2, ..., \tilde{r}^M\}$ and $\tilde{R}' = \{\tilde{r'}^1, \tilde{r'}^2, ..., \tilde{r'}^M\}$ are obtained after the multi-anchor module and time aggregation module with parameter sharing. Finally we aggregate $\tilde{R}$ and $\tilde{R}'$ with self-attention as follow:

$$\hat{r}^i = s^i([\tilde{r}^i, \tilde{r'}^i]) \odot \tilde{r}^i + (1 - s^i([\tilde{r}^i, \tilde{r'}^i])) \odot \tilde{r'}^i, \tag{7}$$

where $\odot$ is the Hadamard product, $[\cdot]$ is a operation concatenating vectors along the last dimension, and $s^i(\cdot)$ is the switch function which is implemented as a fully-connected layer followed by a sigmoid function. The switch function controls the fusion between BoI feature sequences of different scales. The output of this module $\hat{R} = [\hat{r}^1, \hat{r}^2, ..., \hat{r}^M]$ is served as the final user representation.

### 3.2.4 Contrastive Learning Module

As mentioned in 3.1.1, contrastive loss is also used to learn user representation in this network. In order to obtain a unified vector for contrastive learning, a nonlinear projection head is added, i.e. $h(\cdot)$, as used in SimCLR Chen et al. (2020). The function $h(\cdot)$ is computed as

$$\begin{aligned} v &= ReLU([W_1^{h_1}\hat{r}^1, W_2^{h_1}\hat{r}^2, ..., W_M^{h_1}\hat{r}^M]) \\ h(v) &= W^{h_2} ReLU(v + mix(v, W^{h_3})). \end{aligned} \tag{8}$$

The function $mix(v, W^{h_3})$ is to allow interaction between different preferences, and is implemented as

$$mix(v, W^{h_3}) = r_2(W^{h_3} r_1(v)), \tag{9}$$

where $r_1(\cdot)$ reshapes $v$ to a matrix of size $M \times H$, and $r_2(\cdot)$ reshapes $W^{h_3} r_1(v)$ to a vector. $W_i^{h_1} \in \mathbb{R}^{H \times H}$, $W^{h_2} \in \mathbb{R}^{MH \times MH}$, and $W^{h_3} \in \mathbb{R}^{M \times M}$ are trainable parameters.

Two behavior sequences drawn from the same user are treated as a positive pair, and behavior sequences from other users are negatives. All sequences are continuous sub-sequences randomly sampled from the whole life time $\boldsymbol{T}$. Finally, the contrastive loss is defined the same as equation (3).

# 4 EXPERIMENTS

In this section, we present our experiments in detail.

## 4.1 DATASETS

Model comparisons were conducted on a public datatset and an industrial dataset. Table 1 shows the statistics of two datasets. The public Amazon dataset Ni et al. (2019) contains product reviews and metadata like product titles and categories from Amazon. For each user, the reviewed product titles constitute a sequence of review behaviors. The dataset for self-supervised user modeling on Amazon $D_{pt}^A$ was constructed by collecting users who have behaviors between 1997-01 and 2017-12, while taking behaviors occurred between 2018-01 and 2018-10 to form datasets $\{D_{ds}^A(t)\}_t$ ($t$ refers to tasks) for downstream tasks. Industrial dataset was collected from visit/order logs on a popular e-commerce platform. We constructed a training dataset $D_{pt}^I$ for self-supervised user modeling by randomly sampling hundred million of users who have behaviors between 2016-06 and 2021-05, while taking behaviors occurred between 2021-07 and 2021-08 to form datasets $\{D_{ds}^I(t)\}_t$.

Table 1: Statistics of the datasets. $Tr(\cdot)$ indicates the truncation threshold.

| Dataset | $|U|$ | $Max(|\boldsymbol{S}|)$ | $Tr(|\boldsymbol{x}|)$ | $|V|$ |
|---|---|---|---|---|
| Amazon | 43,531,850 | 13,122 | 35 | 103,581 |
| Industrial | 214,317,285 | 1,952,546 | 24 | 178,422 |

## 4.2 DOWNSTREAM TASKS

Two kinds of downstream tasks were used to evaluate our method: category preference identification and user profiling prediction. In the experiments, we randomly selected 80% of the samples from $D_{ds}^A(t)/D_{ds}^I(t)$ for task $t$ to train downstream models and the rest for performance validation.

Category preference identification refers to the task of predicting whether users have short-term preferences in the target category. For Amazon dataset, four categories were included: 'Books of Literature', 'Games of Sports', 'Outdoor-hunting', and 'Musical Instrument'. For the industrial dataset, we considered three categories including clothing, shoe and coffee for evaluation.

User profiling prediction aims to identify user aspects such as gender and age. We only conducted experiments on industrial dataset. Two tasks were involved: (1) user age classification task (age is divided into 6 classes) which predicts the age ranges of users, and (2) baby age classification task (7 class). In both tasks, the ground-truth labels came from an online questionnaire.

## 4.3 COMPETITORS

We compared the proposed LURM model against a rich set of user representation learning methods, including TF-IDF, Doc2Vec, TextCNN, HAN, PTUM and SUMN. TF-IDF and Doc2Vec view the whole user behavior sequence as a document. TF-IDF generates a sparse high-dimensional vector, while Doc2Vec learns to represent the document by a dense vector. We also compared LURM with two methods named TextCNN and HAN which learn task-specific user representations and classifiers in a supervised manner. Finally, we compared LURM with two newly self-supervised methods for user modeling mentioned before, PTUM and SUMN.

## 4.4 TRAINING DETAILS

For item embedding learning, the number of words in each item $\boldsymbol{x}$ was truncated. The principle of setting truncation threshold is that 95% data values can be covered by the threshold. In experiments, word2vec and average pooling were used to obtain item embedding on Amazon dataset, while the method described in 3.1.1 was used on the industrial dataset. The entire behavior sequences were

input into SMEN because the length of input sequence only depends on the time interval. On Amazon dataset, TF-IDF and Doc2Vec also took the entire behavior sequence as input. Meanwhile, the length of input of TextCNN, HAN, Doc2Vec, SUMN and PUTM was limited to 50 due to memory limitation. On industrial dataset, all competitors used data of two months.

We set the latent dimensions of word/item embedding and all hidden layers to 128 in LURM. In item embedding module, the window size $\beta$ was set to 5 days, and the temperature $\tau$ was 0.1. In order to retain details as many as possible, the size of the 'interest' vocabulary was set to $10^5$, and the number of anchors $M$ was set to 10 in our experiments. So the final user embedding dimension was 1280. In addition, the input of month-granularity and year-granularity were used. For all user representation models, the Adam optimizer with a learning rate of 0.001, and a batch size of 256 was used for training. The hyper-parameters of the supervised competitors were tuned on the validation set. For downstream tasks, a simple MLP classifiers was applied after the derived user representation. Note that LURM, SUMN and PUTM were all not fine-tuned for downstream tasks in our experiments.

## 4.5 RESULTS

Table 2 shows the comparison of category preference identification on the public Amazon dataset. The last two rows show the results of our models under different settings. The first one uses the high-dimensional output of BoI module as input in the downstream tasks, while the second uses the output of LURM. It can be seen that LURM consistently outperforms other unsupervised methods, e.g., about 3.23%/1.49% average improvements than SUMN and 4.41%/2.09% average improvements than PUTM in terms of AUC and ACC respectively. Even compared with supervised methods, LURM outperforms TextCNN and HAN about 8.26%/3.47% and 3.82%/2.04% respectively. In particular, it can be observed that LURM has a significant improvement over BoI.

Table 3 shows the comparison of two user profiling prediction tasks and three category preference identification tasks on the industrial dataset. In this table, BoI and LURM using 2-month input are compared. When using the same input, LURM(short) achieved better results than other methods on both user profiling prediction tasks and category preference identification tasks. When using longer input, we can obtain more improvements, e.g., about 15.55%/17.55% average improvement on user profiling prediction tasks, and 4.19%/2.91% average improvement on category preference identification tasks compared with SUMN.

Table 2: Comparison in terms of AUC(%)/ACC(%) on the Amazon dataset. Several category preference identification tasks are evaluated. The best score is bold.

| Method | Books of Literature | Games of Sports | Outdoor-hunting | Musical Instrument |
|---|---|---|---|---|
| TextCNN | 73.84/72.83 | 61.00/69.92 | 66.06/72.76 | 68.28/71.81 |
| HAN | 79.12/77.49 | 66.62/70.01 | 71.19/73.42 | 70.01/72.12 |
| TF-IDF | 78.62/77.21 | 65.69/68.69 | 71.90/72.19 | 69.16/68.62 |
| Doc2Vec | 71.21/68.26 | 66.75/66.85 | 71.58/66.36 | 70.29/67.37 |
| PTUM | 78.66/77.91 | 66.06/68.97 | 70.34/73.01 | 69.50/72.92 |
| SUMN | 79.57/77.73 | 66.83/70.63 | 72.27/73.44 | 70.61/73.43 |
| BoI | 76.29/75.75 | 59.34/67.58 | 67.80/71.28 | 66.36/69.88 |
| LURM | **82.55/79.13** | **68.02/72.43** | **77.59/75.19** | **74.04/74.43** |

## 4.6 ABLATION STUDIES AND DISCUSSION

From Table 2 and Table 3, we have observed that LURM has achieved better performance compared to other self-supervised and supervised methods. Its success can be attributed to lifelong user behavior modeling and lossless dimensionality reduction of SMEN.

Fig. 3 shows the performance of LURM on several downstream tasks of industrial dataset when using inputs of different lengths. It can be seen that the richer the behavior, the better the performance on user profiling prediction tasks. Compared to using only behaviors of two months, an average improvement of 9.75%/16.03% can be achieved when using behaviors of 5 years on these tasks. While the benefits of using richer behavioral data become small for category preference identification tasks. This is because these tasks are more focused on the short-term interests of users, and it can be observed that the best performance can be obtained when input with a length of one year is used on these tasks. We also verified the necessity of multi-scale aggregation module. The

third and fourth rows of Table 4 show the results of LURM with input of monthly/yearly granularity only, respectively. It can be seen that using multi-scale aggregation module can achieve significant improvements on user profiling prediction tasks, while the benefits disappear on category preference tasks. The reason is the same as explained in Fig. 3, that is, there is no need to model representation with extremely long user behaviors for category preference identification tasks. The last two rows of Table 4 show the results of LURM with 1 anchor (LURM(1anchor-128)), and 1 anchor with 1280-dimensional output (LURM(1anchor-1280)). Taking BoI as a benchmark, it can be seen that LURM achieves almost lossless dimensionality reduction compared to LURM(1anchor-128) and LURM(1anchor-1280), which verifies the effectiveness of the multi-anchor module.

Table 3: Comparison in terms of AUC/ACC on Industrial dataset. Two profiling prediction tasks and three category preference identification tasks are evaluated. The best score is bold.

| Method | Age | Baby Age | Clothing | Shoe | Coffee |
|---|---|---|---|---|---|
| TextCNN | 86.40/61.02 | 72.19/66.01 | 76.64/76.13 | 84.33/80.24 | 80.49/79.46 |
| TF-IDF | 87.73/61.75 | 73.08/67.26 | 77.61/78.01 | 85.04/81.15 | 81.75/80.78 |
| SUMN | 85.35/60.61 | 74.20/67.63 | 73.86/74.16 | 83.47/79.63 | 79.89/79.42 |
| BoI(short) | 89.57/63.99 | 82.23/70.22 | 78.93/78.85 | 85.53/81.68 | 82.61/81.41 |
| LURM(short) | 88.48/61.99 | 82.71/69.29 | 78.37/78.57 | 85.15/81.18 | 81.94/80.78 |
| BoI | **96.48/80.19** | 93.29/83.57 | **80.83/79.33** | **86.69/82.16** | **83.78/81.74** |
| LURM | 96.09/78.43 | **94.59/84.91** | 80.37/79.10 | 86.30/81.65 | 83.11/81.18 |

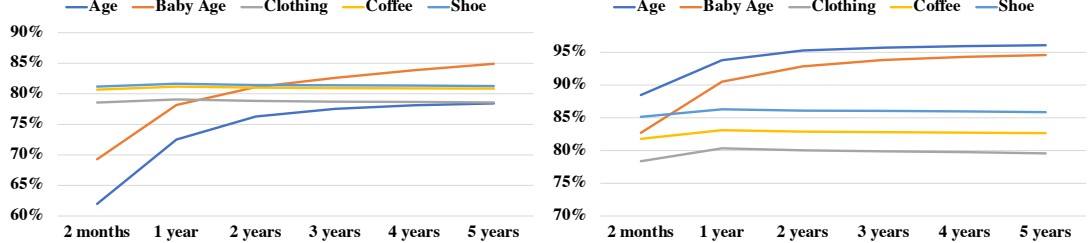

Figure 3: Comparison of LURM with inputs of different lengths on several downstream tasks. The two figures show the results in terms of AUC and ACC respectively.

Table 4: Comparison of LURM with different configurations on industrial dataset.

| Method | Age | Baby Age | Clothing | Shoe | Coffee |
|---|---|---|---|---|---|
| BoI | 96.48/80.19 | 93.29/83.57 | 80.83/79.33 | 86.69/82.16 | 83.78/81.74 |
| LURM | 96.09/78.43 | 94.59/84.91 | 80.37/79.10 | 86.30/81.65 | 83.11/81.18 |
| LURM(monthly) | 95.82/77.54 | 93.19/82.17 | 80.30/79.02 | 86.26/81.61 | 83.08/81.14 |
| LURM(yearly) | 95.61/77.19 | 92.80/80.93 | 80.24/78.88 | 86.17/81.36 | 82.66/80.78 |
| LURM(1anchor-128) | 93.98/72.78 | 86.13/71.22 | 78.93/78.27 | 85.80/81.11 | 81.98/80.39 |
| LURM(1anchor-1280) | 94.35/73.71 | 89.21/75.35 | 79.38/78.54 | 86.07/81.32 | 82.34/80.59 |

In addition, we also observed that the gap between BoI and LURM is very different on the two datasets. It may be that the item embedding we learned is better on the industrial dataset and then LURM achieved similar results compared with BoI, indicating the ability of SMEN to compress representation losslessly. On the Amazon dataset, the item embedding is worse. And It can be seen that the representation was improved through the further learning of SMEN with self-supervised tasks and the ability to encode temporal variation.

## 5 CONCLUSION

In this work, a novel framework named LURM is proposed to model lifelong user behaviors with any length. With the ability to model lifelong user behaviors, our method shows promising results on different downstream tasks and datasets. Although our method has made some progress, there is still space for improvement. In the future research work, we will consider more types of tasks; input data in different modalities, such as images, video, and audio; more dedicated network architecture and so on.

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
