# OpenReview forum: "Learning Universal User Representations via Self-Supervised Lifelong Behaviors Modeling"
_ICLR.cc/2022/Conference — ICLR 2022 Submitted_

### Official Review · Reviewer_Rubm · 2021-10-20

**Correctness:** 3
**Technical Novelty And Significance:** 3
**Empirical Novelty And Significance:** 2
**Recommendation:** 5
**Confidence:** 4

**Main Review:**

Strengths:

--Previous work cannot model universal user representation based on lifelong sequences of user behavior since registration and challenging to make full use of highly long user behavior sequences accumulated in Internet service platforms.

-- This work proposes a universal Lifelong User Representation Model to encode user behaviors in any period with a sparse vector with super-high dimension and map these user features to user representations with almost lossless dimensionality reduction. In this way, the model could easily learn different aspects of user preferences in extremely long sequences and generalize to other downstream tasks.

Weaknesses:

-- The structure of related works is not clear enough. At the end of universal user modeling, this work focuses on the conclusion that previous work dedicated to learning universal user representations cannot leverage information from lifelong user behaviors instead of their generalization ability. The same problem also appeared at the end of lifelong user modeling.

-- In a multi-scale aggregation module, this work captures diverse patterns from user behavior by aggregating several representations generated at different time granularities. However, this method could only use information intra-corresponding representation of every time granularity and cannot explore the information inter different time granularities.

-- The explanation of the considerable gap between BoI and LURM on the two datasets is unconvincing. On the industrial dataset, the performance of BOI is better than LURM, which proved that the design of SMEN cannot bring benefits for BOI.

-- In different downstream tasks, the benefits on category preference identification tasks are tiny, even using behaviors of 5 years on this task. There is no explanation about this phenomenon, and it proves BOI cannot learn much preference information about user behaviors on different categories.

-- This work did not design experiments about multi-scale aggregation modules to prove the effectiveness of process representations of different time granularities.

-- There are some format problems in the paper, such as no period at the end of the first paragraph of the introduction.

**Summary Of The Paper:**

This work proposes a universal Lifelong User Representation Model to encode user behaviors in any period with a sparse vector with super-high dimension and map these user features to user representations with almost lossless dimensionality reduction. In this way, the model could easily learn different aspects of user preferences in extremely long sequences and generalize to other downstream tasks.

**Summary Of The Review:**

This work proposes a universal Lifelong User Representation Model to learn different aspects of user preferences in extremely long sequences and easily generalize to other downstream tasks by the self-supervised method.

However, the experimental design of this work needs to be improved, and more explicability needs to be provided about crucial modules and abnormal experimental phenomena, which have more enlightening significance for the community.

---

> ### Author Response · Authors · 2021-11-21
> **Response to Reviewer Rubm(3/3)**
>
> **Q6:** *There are some format problems in the paper, such as no period at the end of the first paragraph of the introduction.*
>
> **A6:** Thanks for pointing out the problem, we have revised these format errors in the latest version.
>
> **Q7:** *Important experiments about an explanation of key modules and abnormal experiment phenomena are required in this work, reducing the interpretability of the work.*
>
> **A7:** We have presented almost all important experiments about an explanation of key modules in Section 4. The benefits of SMEN can be seen from Table 2 and Table 3. The importance of the length of user behaviors is shown in Table 3 and Fig. 3. The necessity of multi-scale aggregation module and multi-anchor module can be seen from Table 4. We have added a more detailed explanation of abnormal experiment phenomena in the last paragraph of section 4.6.

---

> ### Author Response · Authors · 2021-11-21
> **Response to Reviewer Rubm(2/3)**
>
> **Q4:** *The explanation of the considerable gap between BoI and LURM on the two datasets is unconvincing. On the industrial dataset, the performance of BOI is better than LURM, which proved that the design of SMEN cannot bring benefits for BOI.*
>
> **A4:** First of all, the purpose of SMEN design is not to further improve on the basis of BoI, but to solve the two flaws of BoI that BoI is super-high dimensional and dose not model temporal variations.
>
> 1) SMEN reduces the BoI sequence with extremely high dimensionality (e.g. $10^5$) to low dimensionality (e.g. 1280), which is easy to use for downstream tasks. Such a large degree of dimension reduction will definitely lead to performance degradation, so we design a series of modules in SMEN to avoid loss as much as possible.
>
> 2) SMEN models variations in time in the form of BoI sequence instead of one BoI which statistics the lifelong information without difference in time. At this point,  SMEN will bring performance improvements.
>
> In summary, the performance of SMEN will increase or decrease compared with BoI depends on the dataset and the task. For such heavy dimensionality reduction, we can accept a certain degree of performance degradation. In our experiments on industrial datasets, we have also seen that the performance of LURM is still much better than other user modeling methods although SMEN brings a little loss due to dimensionality reduction.
>
> The reason for the gap between BoI and LURM on the two datasets may be the different ways we generate item embeddings. For the industrial dataset, we believe that the items bought and visited within several days are related as positive pairs. And we train an encoder with contrastive learning to generate item embedding. While for the amazon dataset, we can't construct such positive pairs, so a pretrained word2vec was used to generate the item embedding by simply calculating the mean of embeddings of all words in title. This may be the reason why BoI and LURM work differently on the two datasets. On the industrial dataset, we think that the item embedding trained with self-supervised tasks will enrich the information represented by BoI. So SMEN with additional ability to encode temporal variation cannot further improve the performance. While on the amazon dataset, the BoI embedding is much worse, the final representation can be improved by SMEN with self-supervised tasks and the ability to encode temporal variation.
>
> **Q5:** *In different downstream tasks, the benefits on category preference identification tasks are tiny, even using behaviors of 5 years on this task. There is no explanation about this phenomenon, and it proves BOI cannot learn much preference information about user behaviors on different categories.*
>
> **A5:** Firstly, compared with two state-of-the-art universal user representation models, our method achives better results on all category preference identification tasks. In Table 2, it can be seen that LURM obtains about 3.23%/1.49% average improvements than SUMN and 4.41%/2.09% average improvements than PUTM in terms of AUC and ACC respectively on the public amazon dataset. And on the industrial dataset,  LURM obtains about 4.19%/2.91% average improvements than SUMN on category preference identification tasks, which is shown in Table 3. Even compared with supervised methods: TextCNN or HAN, LURM outperforms TextCNN and HAN about 8.26%/3.47% and 3.82%/2.04% respectively on the public Amazon dataset.
>
> Secondly, we would like to explain that category preference identification tasks focuse more on short-term interest of users, so it will not improve continuously with user behaviors grow in time. For example, what you liked 5 years ago have little effect on what you buy in the next month. So as we state in the second paragraphs of section 4.6 that using 1 year user behaviors is enough to predict user's short-term category preference.
>
> Finally, we want to emphasize that the proposal of our lifelong user modeling method was originally designed for user profiling prediction tasks focuse more on long-term interest of users, and the good performance on category preference tasks is an extra surprise. It shows that our entire algorithm framework is very suitable for both short-term and long term interest preference prediction tasks.

---

> ### Author Response · Authors · 2021-11-21
> **Response to Reviewer Rubm(1/3)**
>
> Thanks for your review. We summarize your review as following points and provide discussions and explanations.
>
> **Q1:** *The structure of related works is not clear enough. At the end of universal user modeling, this work focuses on the conclusion that previous work dedicated to learning universal user representations cannot leverage information from lifelong user behaviors instead of their generalization ability. The same problem also appeared at the end of lifelong user modeling.*
>
> **A1:** We are concerned about both leveraging information from lifelong user behaviors and generalization ability.
>
> In the part of `user modeling',  previous work with respect to user universal representations modeling already have good generalization capabilities, so we does not intend to explore this aspect further in our work. More importantly, we observed that existing methods are severely limited by the length of user behaviors, resulting in limited performance in each downstream task. So here we emphasize the ability of lifelong user modeling.
>
> In the part of `lifelong user modeling', we states that those CTR models have the ability to model lifelong user behaviors, but they are task-specific. So we have better generalization ability than them.
>
> Overall, our work performs well in both aspect of generalization ability and lifelong modeling. Additionally, we have performed on 4 downstream tasks on amazon dataset and on 5 downstream tasks on industrial dataset which are always used to test the generalization ability of universal user representation. Since our method outperforms other methods on all the tasks, we stressed the ability of leveraging information from lifelong user behaviors instead of the generalization ability that all the `universal user representation' model should have.
>
> **Q2:** *Multi-scale aggregation module could only use information intra-corresponding representation of every time granularity and cannot explore the information inter different time granularities.*
>
> **A2:** We aggregate information inter different time granularities with self-attention. It is shown in equation (7).
>
> **Q3:** *This work did not design experiments about multi-scale aggregation modules to prove the effectiveness of process representations of different time granularities.*
>
> **A3:** We have already done ablation experiments about multi-scale aggregation modules and the result is shown in Table 4. The second, third and fourth rows of Table 4 show the results of LURM with multi-scale input including monthly granularity and yearly granularity, monthly single-scale input and yearly single-scale input respectively. It can be seen that multi-scale aggregation module can obtain different degrees of improvement on different downstream tasks, especially on user profiling prediction tasks. Take the baby age task as an example,  LURM with multi-scale aggregation modules brings an increase of 1.4%/2.74%(AUC/ACC) compared to LURM with monthly single-scale input, and brings an increase of 1.79%/3.98%(AUC/ACC) compared to LURM with yearly single-scale input.

---

### Official Review · Reviewer_2k8K · 2021-10-31

**Correctness:** 3
**Technical Novelty And Significance:** 3
**Empirical Novelty And Significance:** 2
**Recommendation:** 5
**Confidence:** 4

**Main Review:**

Strengths:

1. This paper is easy to understand.
2. The problem studied in this paper is important.
3. The proposed multi-scale aggregation method is interesting.

Weaknesses:

1. The contribution of this paper can be somewhat incremental. First, the multi-anchor attention network is also called "multi-aspect attention", which has been widely used in many scenarios [1,2,3]. A very similar contrastive model pertaining approach is also studied in [4]. Thus, the contributions of this paper are somewhat incremental.
2. Discussions and comparisons on universal user modeling methods are insufficient. The authors only include PTUM as the baseline but do not discuss it in Section 2.1. There are also many other methods like [4,5]. They are also not cited nor compared.
3. The design of the item embedding method is problematic. Different from texts that there are rich relations between adjacent words, even adjacent behaviors can be rather random. Thus, it may not be suitable to learn item embeddings by regarding items as words and user behavior sequences as sentences. There are many methods for item embedding learning such as PinSage [6].
4. Compared baselines are also insufficient. As mentioned in the second point, there are many user model pretraining methods that are not compared. In addition, the compared basic user modeling techniques are also rather weak (e.g., CNN, HAN, Doc2Vec and TF-IDF).
5. Some details are unclear. PTUM and PeterRec require fine-tuning the user model in downstream tasks. However, in this work it is not clear whether the pretrained user model is finetuned.
6. The selection of some hyperparameters seems to be arbitrary, e.g., the number of anchors and the window size $\beta$. The authors do not provide experiments to show the influence of several key hyperparameters.
7. The generality of the proposed method is not verified. It is not clear whether the proposed method can be generalized to other types of user models.


Minor:

There are some typos and grammatical issues. For example, "committed to learn" and "Comparisonn".

[1] Event detection using hierarchical multi-aspect attention. WWW 2019.

[2] ARP: Aspect-aware neural review rating prediction. CIKM 2019.

[3] Hybrid graph convolutional networks with multi-head attention for location recommendation. World Wide Web 23.6 (2020): 3125-3151.

[4] Contrastive Pre-training for Sequential Recommendation. arXiv e-prints (2020): arXiv-2010.

[5] Parameter-efficient transfer from sequential behaviors for user modeling and recommendation. SIGIR. 2020.

[6] Graph convolutional neural networks for web-scale recommender systems. KDD 2018.

**Summary Of The Paper:**

This paper introduces a universal user representation learning approach based on self-supervised learning on long-term user behaviors. The authors first represent user behaviors into sparse vectors, and then encode them into multiple vectors to represent users in different aspects. The authors further study using different time aggregation strategies to tradeoff accuracy and computational cost. Experiments on two datasets show the effectiveness of the proposed approach.

**Summary Of The Review:**

This work has some merits but it suffers from several issues that need to be addressed. Thus, my recommendation is a weak rejection.

---

> ### Author Response · Authors · 2021-11-21
> **Response to Reviewer 2k8K(2/2)**
>
> **Q4:** *Compared baselines are also insufficient. As mentioned in the second point, there are many user model pretraining methods that are not compared. In addition, the compared basic user modeling techniques are also rather weak (e.g., CNN, HAN, Doc2Vec and TF-IDF).*
>
> **A4:** To make the comparison more convicing, we compared our method with two newly methods of universal user modeling: PTUM [8] and SUMN [7],  two supervised methods TextCNN and HAN, and two classical methods TF-IDF and Doc2Vec. We think the compared baselines are sufficient.
>
> **Q5:** *Some details are unclear. PTUM and PeterRec require fine-tuning the user model in downstream tasks. However, in this work it is not clear whether the pretrained user model is finetuned.*
>
> **A5:** PTUM、SUMN and our LURM all can be fine-tuned in downstream tasks, but it go against our original intention of learning universal representation. In order to be consistent with our method, we did not fine-tune PTUM in our experiments. PeterRec was not performed. We have done some changes to make it clearer in the paper.
>
> **Q6:** *The selection of some hyperparameters seems to be arbitrary, e.g., the number of anchors and the window size . The authors do not provide experiments to show the influence of several key hyperparameters.*
>
> **A6:** Thanks for the suggestion. Due to the limitation of time and the length of the article, we did not write all experiments down except some more important studies. More detailed results will be appended to appendix later. We have shown the benefits of SMEN in Table 2 and Table 3, the importance of the length of user behaviors in Table 3 and Fig. 3, the necessity of multi-scale aggregation module and multi-anchor module in Table 4.
>
> **Q7:** *The generality of the proposed method is not verified. It is not clear whether the proposed method can be generalized to other types of user models.*
>
> **A7:** Our method is based on the modeling of user behavior sequences. As long as there are behaviors, such as purchased items, visited places,  read articles and so on, we can learn the representations of these objects just like learning item embedding. After obtaining the embedding of these objects, subsequent modeling is the same.
>
> **Q8:** *There are some typos and grammatical issues.*
>
> **A8:** Thanks for pointing out the problem, we have  carefully checked these grammatical issues in the latest version.
>
> [1] Event detection using hierarchical multi-aspect attention. WWW 2019.
>
> [2] ARP: Aspect-aware neural review rating prediction. CIKM 2019.
>
> [3] Hybrid graph convolutional networks with multi-head attention for location recommendation. World Wide Web 23.6 (2020): 3125-3151.
>
> [4] Contrastive Pre-training for Sequential Recommendation. arXiv e-prints (2020): arXiv-2010.
>
> [5] Parameter-efficient transfer from sequential behaviors for user modeling and recommendation. SIGIR. 2020.
>
> [6] Graph convolutional neural networks for web-scale recommender systems. KDD 2018.
>
> [7] Exploiting Behavioral Consistence for Universal User Representation. AAAI 2021.
>
> [8] Ptum: Pre-training user model from unlabeled user behaviors via self-supervision. arXiv preprint
> arXiv:2010.01494, 2020.

---

> ### Author Response · Authors · 2021-11-21
> **Response to Reviewer 2k8K(1/2)**
>
> We appreciate the reviewer for the constructive comments. We summarize the review as following points and provide discussions and explanations.
>
> **Q1:**  *The contribution of this paper can be somewhat incremental. First, the multi-anchor attention network is also called `multi-aspect attention', which has been widely used in many scenarios [1,2,3]. A very similar contrastive model pertaining approach is also studied in [4]. Thus, the contributions of this paper are somewhat incremental.*
>
> **A1:** About the novelty of our method, please refer to our reply "**General response to 'the novelty of our work'**". Then we reply to questions you mentioned in detail：
>
> 1) Firstly, we want to emphasize that the multi-anchor module in our work is quite different from the attention used in [1,2,3]. Our multi-anchor attention maps each BoI feature to different anchors, and the representation corresponding to each anchor is the aggregation of all elements of the BoI related to the anchor and is computed as ${r}^i=ReLU(\sum_j^D\alpha_{ij}b_jW_j^e)$, where $\alpha_{ij}=\frac{\exp(f(W_i^a, W_j^e))}{\sum_{\textcolor{red}{l}}\exp(f(\textcolor{red}{W_l^a}, W_j^e))}$, $f$ is a learned function to measure the relation between i-th anchor $W_i^a$ and j-th BoI element $W_j^e$. While $\alpha_{ij}$ is computed as $\frac{\exp(f(W_i^a, W_j^e))}{\sum_{\textcolor{red}{k}}\exp(f(W_i^a, \textcolor{red}{W_k^e}))}$ in multi-head attention, which models the interaction between target anchor and the BoI elements. In multi-head attention, the sum of weights of all elements in the BoI related to each anchor is always 1. But in fact, each user has different interests, and the degree of preference for different interests is also different. Therefore, we believe that a more reasonable way to assign weights is like what multi-anchor do. We have done some experiments to verify that the implementation of multi-anchor attention has more improvements. But due to time limitations, we did not write it in the article. We will add more experimental results in the appendix in the future.
>
> 2) Secondly, even though contrastive learning is also used in [4], there are still some differences. The contrastive learning in [4] takes different views of the same sequence as positive pairs. In our model, in BoI, two items construct a positive pair if they are drawn from the same user behavior sequence in time interval $\beta$. And in SMEN, two behavior sequences drawn from the same user in different time period are treated as a positive pair.
>
> 3) It's worth noting that contrastive learning is a part of our overall model, the main contribution of this work is that we proposed a framework that can model lifelong user behaviors of any length. Experiments show that the richer the behavior, the better the performance. Especially, this main contribution of the ability of lifelong behavior modeling is not available in any other universal user representation models.
> **Q2:** *Discussions and comparisons on universal user modeling methods are insufficient. The authors only include PTUM as the baseline but do not discuss it in Section 2.1. There are also many other methods like [4,5]. They are also not cited nor compared.*
>
> **A2:** Thanks for your suggestion. We already have some discussion of PTUM in Section 4.3 and moved the description of PTUM to Section 2.1. In this work, we compared our method with two SOTA methods of universal user modeling PTUM [8] and SUMN [7], which are published last year. The methods presented in [4] and [5] are designed for recommendation Specifically. Because PTUM and SUMN are more similar to our method, we compared them rather than [4] and [5].
>
> **Q3:** *The design of the item embedding method is problematic. Different from texts that there are rich relations between adjacent words, even adjacent behaviors can be rather random. Thus, it may not be suitable to learn item embeddings by regarding items as words and user behavior sequences as sentences. There are many methods for item embedding learning such as PinSage [6].*
>
> **A3:** In order to make the positive pair more reliable, two items are taken as a positive pair if they are drawn from the same user behavior sequence and the time interval between the occurrence of these two items is less than $\beta$. If $\beta$ is larger, more invariance would be learned. Additionally, two different items generated by the same user may contain the same preferences. So we think our way to generate pasitive pairs is resonable, and experiments show the effectiveness of out method. Of course, other methods for item embedding learning can also be implanted in our framework. The specific implementation of the item embedding module is flexible.

---

### Official Review · Reviewer_FiJs · 2021-11-02

**Correctness:** 2
**Technical Novelty And Significance:** 2
**Empirical Novelty And Significance:** 2
**Recommendation:** 5
**Confidence:** 3

**Main Review:**

[Advantages]
- Modeling users and user behaviors are essential, especially for web-based applications. The problem dealt with in this paper will be a promising application of representation learning based on neural networks.
- The structure of the proposed method is reasonable, especially for e-commerce applications.

[Drawbacks]
- I could not understand the novelty of the proposed method. The bag-of-features representation seems to be standard in several research fields such as computer vision, natural language processing, and data mining. The "multi-anchor module" seems to be almost the same as the multi-head attention module that is famous in the transformer. The time aggregation module does not contain a special trick. According to the introduction of this paper, the novelty of the proposed method is the ability for modeling life-long behaviors. However, I could not find any special techniques for handling life-long sequences. Please clearly state the technical novelty of the proposed method.
- The proposed method for modeling user behaviors dismisses both microscopic and macroscopic time-series characteristics, which would be significant for tracing user interest. More specifically, bag-of-interest representations remove microscopic characteristics, and the time aggregation module erases macroscopic features. If the authors believe that I have some misunderstanding, please justify it.
- When obtaining stable user representations, the knowledge of life-long learning will be required, since (1) users might have several completely different interests in their minds, (2) each interest lasts only a short term, especially in e-commerce applications, and (3) interests containing old behavior sequences might come back suddenly. This implies that almost all the information containing behavior sequences should be kept in user representations and this problem is exactly in the context of life-long learning. Please perform bibliographic surveys for life-long learning, describe the critical problem for the previous work, and present the novelty and advantage(s)  of the proposed method against the previous work.
- The downstream tasks in the experimental evaluations seem to be too easy and not suitable for evaluating the ability to handle long behavior sequences.

**Summary Of The Paper:**

This paper deals with the problem of obtaining user representations from behavior sequences and proposes a method introducing the idea of a "bag of features" and multi-head attention. More specifically, a behavior sequence is expressed by a sequence of multi-dimensional vectors and a fixed-length segment of the sequence is converted to a histogram, where each bin corresponds to a cluster of vectors. Each is then fed to a multi-head attention module to obtain a segment representation and all the segment representations are aggregated into a single vector through a time aggregation module. Experimental evaluations for several downstream tasks demonstrate the effectiveness of the proposed method against several simple baselines.

**Summary Of The Review:**

I could not find any positive aspects for accepting this paper to the premier conference in the field of representation learning. Both the methodology and documentation need to be improved for acceptance.

[Post-rebuttal comments]
After reading through the paper and author rebuttals, one of the reasons for misunderstanding among us is a gap in the target time duration of "microscopic" and "macroscopic". My understanding is
- "Microscopic" user behaviors correspond to short sequences of individual actions during single sessions, which last at most a day.
- From this viewpoint, the proposed BoI representation removes "microscopic" information that will be useful for the next user action.
- Monthly and yearly user behaviors might be "microscopic" since a month or a year is shorter than their lifetime.
- However, a month is sufficiently long to capture whole the interests of users, which implies that monthly behaviors are "macroscopic".

I understand that the authors try to model "users", not "user behaviors", as implied by the downstream tasks. I also understand that "microscopic" (by my definition) sequence modeling is not required and only "macroscopic" (again by my definition) user behaviors are only of interest.

I have just revised my evaluations according to the above discussion.

---

> ### Author Response · Authors · 2021-11-21
> **Response to Reviewer FiJs(2/2)**
>
> **Q2:** *The proposed method for modeling user behaviors dismisses both microscopic and macroscopic time-series characteristics, which would be significant for tracing user interest. More specifically, bag-of-interest representations remove microscopic characteristics, and the time aggregation module erases macroscopic features. If the authors believe that I have some misunderstanding, please justify it.*
>
> **A2:**
>
> 1) We agree with you that BoI loses some microscopic characteristics. But as we mentioned in the paper that BoI can be performed at different granularities, for instance, monthly and yearly. The finer the granularity, the fewer details will be lost.
>
> 2) However, time aggregation module does not erase any macroscopic features in our opinion. In this work, GRU (a type of RNN) is used to model macroscopic time-series characteristics. It is a basic knowledge that RNN can model variations in time series.
>
> 3) In our method, we use BoI to encode behaviors in any time peroid, and then use SMEN to model macroscopic time-series characteristics. Although PUTM and SUMN are methods that can encode microscopic time-series characteristics,  we still find that LURM(short) can achieve better results even on three category preference identification tasks which need microscopic time-series characteristics on the industrial dataset in Table 3. This proves that the design of our method is effective.
>
> **Q3:** *When obtaining stable user representations, the knowledge of life-long learning will be required, since (1) users might have several completely different interests in their minds, (2) each interest lasts only a short term, especially in e-commerce applications, and (3) interests containing old behavior sequences might come back suddenly. This implies that almost all the information containing behavior sequences should be kept in user representations and this problem is exactly in the context of life-long learning. Please perform bibliographic surveys for life-long learning, describe the critical problem for the previous work, and present the novelty and advantage(s) of the proposed method against the previous work.*
>
> **A3:**
>
> 1) For the first opinion, we agree with you that user have several different interests and actually we already designed a multi-anchor module to handle it.
>
> 2) For the second opinion, we totally doubt it. Why do you think each interest lasts only a short term? There are also many interests last long term, such as sports preference, electronics preferences, cars preferences and so on. And there are also many user profiling prediction tasks which need to catch user's long-term interests, such as age, baby age and so on. These are all mentioned in our paper repeatly.
>
> 3) The third opinion actually is one of the reasons for the need of lifelong user behavior modeling. But we can't agree with the opinion that life-long (continual) learning is the only way to learn representations that contain all the information of behavior sequences. In fact, no matter what method is used, it will definitely lead to a certain degree of information loss. Our multi-anchor module and multi-scale aggregation module introduced have the ability to alleviate catastrophic forgetting.
>
> Overall, we insist that life-long learning is not the only way to solve the problems above. And our method has considered these points that you care about.
>
> **Q4:**  *The downstream tasks in the experimental evaluations seem to be too easy and not suitable for evaluating the ability to handle long behavior sequences.*
>
> **A4:**
>
> 1) In the field of universal user representation, these experiments are always performed to evaluate the methods[1,2]. And in the industry, what we care about most just is the performance of these tasks.
>
> 2) User profiling prediction tasks such as age in our experiments are very suitable for evaluating the ability to handle long behavior sequences. For example, if our model catch the information that the user bought an primary school guidance book 5 years ago, it will help to infer the user's age now is in stage of high school. And our experiments validate that as the sequence increases, the performances in user profiling prediction tasks will get better. It is sufficient to show that these tasks are suitable for evaluating the ability to handle long behavior sequences.
>
> [1] Gu, Jie, et al. `Exploiting Behavioral Consistence for Universal User Representation.' arXiv preprint arXiv:2012.06146 (2020).
>
> [2] Ptum: Pre-training user model from unlabeled user behaviors via self-supervision. arXiv preprint
> arXiv:2010.01494, 2020.

---

> ### Author Response · Authors · 2021-11-21
> **Response to Reviewer FiJs(1/2)**
>
> Thank you for your review, but some of your opinions are unexpected to us. There is a gap between our understandings. We have provided discussions and explanations around your concerns as follows.
>
> **Q1.**  *I could not understand the novelty of the proposed method. The bag-of-features representation seems to be standard in several research fields such as computer vision, natural language processing, and data mining. The 'multi-anchor module' seems to be almost the same as the multi-head attention module that is famous in the transformer. The time aggregation module does not contain a special trick. According to the introduction of this paper, the novelty of the proposed method is the ability for modeling life-long behaviors. However, I could not find any special techniques for handling life-long sequences. Please clearly state the technical novelty of the proposed method.*
>
> **A1:** About the novelty of our method, please refer to our reply "**General response to 'the novelty of our work'**". Then we reply to other questions you mentioned in detail：
>
> 1) First of all, the overall pipeline we proposed is brand new and it performs better than other methods **in the field of universal user representation learning**. The experimental results show that our approach achieves the state-of-the-art. Our innovation is introducing a **framework** to model **lifelong** user behaviors at any length by combining BoI which can encode behaviors in any time period, and SMEN which can obtain compressed user representations with little information loss. To the best of our knowledge, it is the first work to model lifelong universal user representation. If there exists other works, please share with us.
>
> 2) About realizing the ability to model lifelong user behavior, it is mainly through the BoI module. We first split the lifelong behaviors (with a length up to millions) into multiple periods at a certain granularity. Then we obtain the BoI representation of each period (sparse vector with about $10^5$ dimension), and get a BoI sequence (with a length of tens or hundreds). At last, we use SMEN to obtain compressed user representations (dense vector with hundreds or thousands dimension) with little information loss.
>
> 3) Then we want to emphasize that the multi-anchor module in our work is quite different from the multi-head attention. Our multi-anchor attention maps each BoI feature to different anchors, and the representation corresponding to each anchor is the aggregation of all elements of the BoI related to the anchor. And it is computed as ${r}^i=ReLU(\sum_j^D\alpha_{ij}b_jW_j^e)$, where $\alpha_{ij}=\frac{\exp(f(W_i^a, W_j^e))}{\sum_{\textcolor{red}{l}}\exp(f(\textcolor{red}{W_l^a}, W_j^e))}$, $f$ is a learned function to measure the relation between i-th anchor $W_i^a$ and j-th BoI element $W_j^e$. While $\alpha_{ij}$ is computed as $\frac{\exp(f(W_i^a, W_j^e))}{\sum_{\textcolor{red}{k}}\exp(f(W_i^a, \textcolor{red}{W_k^e}))}$ in multi-head attention, which models the interaction between target anchor and the BoI elements. In multi-head attention, the sum of weights of all elements in the BoI related to each anchor is always 1. But in fact, each user has different interests, and the degree of preference for different interests is also different. Therefore, we believe that a more reasonable way to assign weights is like what multi-anchor do. We have done some experiments to verify that the implementation of multi-anchor attention has more improvements. But due to time limitations, we did not write it in the article. We will add more experimental results in the appendix in the future.
>
> 4) Finally, about BoI module/time aggregation module that you questioned, what we want to explain here is: We properly adapted some existing techniques and optimized them for practical scenarios of universal user representation. And the time aggregation module is just a functional module used to model temporal variations. The specific implementation method is open, not the focus of our paper.

---

### Author Response · Authors · 2021-11-21
**General response to 'the novelty of our work'**

We thank the reviewers for their careful and valuable comments. We will explain reviewers' concerns point by point. But before that, we want to emphasize the innovation of our method here.

We want to emphasize that this work presents a pioneering model for learning effective **lifelong universal** user representation. Before this work, we are even not sure whether it is possible to learn universal user representation with massive lifelong behaviors (there could be ten thousands or even millions of behaviors for each user), and whether such lifelong universal representations can benefit the performance on various downstream tasks. To the best of our knowledge, we are the first to propose the concept of  'lifelong universal user representation'.

There are several issues that we want to further explain. First, we aim to generate general-purpose user representations by mining and capturing underlying patterns behind user behaviors. Our method can genereate a group of users that have similar interests (e.g., have the same preference on some category), which can be used as a Recall Engine in the recommender. This is very different from the Rank Engine that usually focuses on mining 'very recent' interests of users to improve the CTR. Moreover, it also explains why we adopt category preference prediction as downstream tasks. All the evaluations are designed based on our real-world application scenarios. Learning satisfactory user representation that can be seamlessly applied to dozens (even hundreds) of downstream tasks without fine-tuning is of course very challenging.

Second, we attempt to improve universal user representation by mining richer information with more historical behaviors. That is why we introduce lifelong behaviors (a given user could have millions of behaviors). In this paper, we show that aggregating a large number of user behaviors in a statistic-like way (e.g., using a bag-of-interest module as we do) could be a standard and promising operation in lifelong representation learning. It is obviously impractical to treat millions of behaviors as a sequence and aggregate them with RNN or transformer directly. Such a statistical model exhibits a good trade-off between effectiveness and efficiency.

In addition, we design a multi-anchor module to capture different aspects of users' interests.We want to emphasize that the multi-anchor module in our work is quite different from the multi-head attention. Our multi-anchor attention maps each BoI feature to different anchors, and the representation corresponding to each anchor is the aggregation of all elements of the BoI related to the anchor. And it is computed as ${r}^i=ReLU(\sum_j^D\alpha_{ij}b_jW_j^e)$, where $\alpha_{ij}=\frac{\exp(f(W_i^a, W_j^e))}{\sum_{\textcolor{red}{l}}\exp(f(\textcolor{red}{W_l^a}, W_j^e))}$, $f$ is a learned function to measure the relation between i-th anchor $W_i^a$ and j-th BoI element $W_j^e$. While $\alpha_{ij}$ is computed as $\frac{\exp(f(W_i^a, W_j^e))}{\sum_{\textcolor{red}{k}}\exp(f(W_i^a, \textcolor{red}{W_k^e}))}$ in multi-head attention, which models the interaction between target anchor and the BoI elements. In multi-head attention, the sum of weights of all elements in the BoI related to each anchor is always 1. But in fact, each user has different interests, and the degree of preference for different interests is also different. Therefore, we believe that a more reasonable way to assign weights is like what multi-anchor do. We have done some experiments to verify that the implementation of multi-anchor attention has more improvements.

Last but not least, we reconstruct the contrastive learning for representation learning by treating two behavior sequences drawn from the same user in different time period as a positive pair. The way of constructing positive pairs enables our model to learn the inherent and stable interests of users.

To sum up, our method is radically different from previous ones. It is not fair that reviewers question the novelty of our work only according to some specific network implementations, like the bag-of-feature or multi-anchor module. The formulation of learning lifelong universal user representation and the insights we provided in addressing such a challenging problem are the actual contributions of this work. And experiments on several benchmark datasets show that our approach outperforms other state-of-the-art self-supervised representation methods on downstream tasks, even on short-term interest-related tasks.

---

### Decision · Program_Chairs · 2022-01-20

**Decision:**

Reject

**Comment:**

Ultimately somewhat below the threshold based on the scores. The reviewers raise issues of the overall contribution, as well as issues with the design/structure of the model/paper and issues with the experiments. While there are some positive aspects, collectively the issues put the paper below the bar for acceptance.